# SparseSkeleton: Dynamic Prefill Sparse Attention by Online Decomposition

## Abstract

Multi-head attention (MHA) and grouped query head attention (GQA) consti­tute essential architectural components of modern large language models (LLMs). Even though attention computations remain relatively inexpensive for small-scale inputs, the computational cost increases quadratically as the input size expands. In long-context scenarios, including tasks such as book-level summarization or code repos analysis, time-to-first-token (TTFT) performance can deteriorate sig­nificantly. Although various studies have improved prefill stage performance by exploiting sparsity structure, sparsity can still be further increased with structure refinements.

In this work, we propose an approximate on-line decomposition of the attention matrix which is able to dynamically identify additional sparsity. The attention matrix is decomposed into three components: a slash component, a vertical com­ponent, and a horizontal component. Each component requires only linear space, thereby enabling more efficient processing compared to the full attention matrix. The decomposition is computed from query and key tokens using a linear-time algorithm. The statistical properties of the decomposition allow generation of the mask by merely selecting elements that exceed a threshold. The threshold itself can be chosen to limit the difference with regular dense attention or to respect a certain time-budget.

We demonstrate that this technique can be directly applied – without requiring retraining – to networks employing standard dense attention mechanisms (MHA, GQA) and RoPE. We show that precision is maintained across the $\infty$Bench and PG-19 benchmarks for Llama-3-8B-Instruct-1048k. Furthermore, we ob­serve substantial increases in sparsity and corresponding speedup compared to previous methods. We halve the number of FLOP relative to State-of-the-Art on one million tokens.

## 1 Introduction

Transformer-based large language models (LLMs) have seen spectacular adoption in all kinds of natural language processing tasks. Some particular tasks, such as book summarization, require the LLM to process a long text, split into a large number of tokens, in a phase known as prefill, before it can even start to respond. Transformer LLMs build an attention map between all possible pairs of tokens. Prevalent attention mechanisms, multi-head attention (MHA) and grouped-query attention (GQA), do this explicitly and thus require quadratic number of operations to complete the map. For long texts this time adds up and becomes a major bottleneck that impedes further adoption.

This problem has been identified and addressed by many works before. One particular strategy, employed by Xiao et al. (2024b); Jiang et al. (2024); Xiao et al. (2025); Lai et al. (2025); Sahni et al. (2025); Gao et al. (2024); Zhang et al.; Peng et al. (2025), is to predict a pattern in the attention map before building it. They exploit the fact that the attention map, effectively a matrix $\boldsymbol{A} \in \mathbb{R}^{N \times N}$, is often very sparse. Many of its elements are so close to zero, that their contribution to the attention computation is negligible. By predicting the pattern, one can construct an attention mask that masks the zero elements in the computation, and thus save considerable time.

However, pattern prediction and mask construction introduce computational overhead. Unfortu­nately, offline recognition of static patterns applicable to any input is not possible as inputs influence

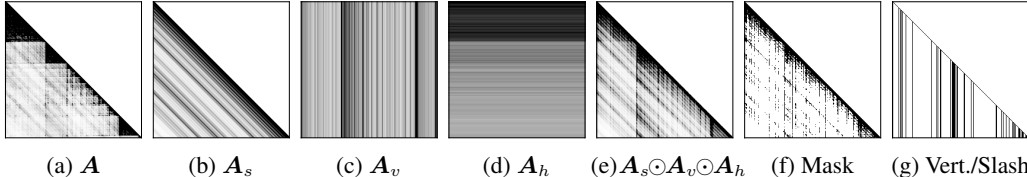

(a) $\boldsymbol{A}$    (b) $\boldsymbol{A}_s$    (c) $\boldsymbol{A}_v$    (d) $\boldsymbol{A}_h$    (e) $\boldsymbol{A}_s{\odot}\boldsymbol{A}_v{\odot}\boldsymbol{A}_h$    (f) Mask    (g) Vert./Slash

Figure 1: An attention matrix (a) from layer 21, head 8 in LLAMA-3-8B-INSTRUCT-1048K together with its decomposition (b-d), the recomposition (e), our final mask (f), and the mask from a vertical/slash method (g). The prompt was a concatenation of several short books from the PG-19 (Rae et al., 2020). This creates the distinct staircase pattern in the attention matrix which is reproduced by our method.

patterns. Hence, this has to be done online and previous methods restrict themselves to a small number of categories, such as $\Lambda$-shape, vertical-slash, block-sparse, and query-aware.

In this paper, we propose to generalize these patterns. In particular, we show a new dynamic (on-line) method to approximate the attention matrix $\boldsymbol{A}$. To the best of our knowledge, this is the first attempt to decompose the attention matrix. As illustrated in Figure 1, this matrix can be decomposed into three components: a slash component $\boldsymbol{A}_s$ with constant diagonals, i.e. Toeplitz, a vertical compo­nent $\boldsymbol{A}_v$ with constant columns, and a horizontal component $\boldsymbol{A}_h$ with constant rows. Equivalently, these components are compactly represented by vectors $\boldsymbol{s}, \boldsymbol{v}, \boldsymbol{h} \in \mathbb{R}^N$ which provide an efficient data-structure for constructing the attention mask. This decomposition captures most relevant emer­gent sparsity patterns in $\boldsymbol{A}$. The decomposition algorithm has linear time complexity which makes it suitable to be used on-line, so that it can dynamically adapt to the sparsity pattern in each prefill phase. Moreover, the distribution of values in $\boldsymbol{s}, \boldsymbol{v}$, and $\boldsymbol{h}$ is well behaved and allows the mask to be generated with only element- or block-wise operations.

Using the LLAMA-3-8B-INSTRUCT-1048K model from Grattafiori et al. (2024), we observe more than 99.5% sparsity on average for inputs of one million tokens. In smaller context, 128K tokens, we obtain 97% sparsity ; and 89% sparsity for 10K tokens. Moreover, through the $\infty$ bench (Zhang et al., 2024), and the PG-19 (Rae et al., 2020) benchmarks we observe that model accuracy is main­tained.

## 2 ATTENTION

Transformer-based LLMs have an attention mechanism that builds and applies an attention map between all possible token pairs in the input. The attention map is in essence a matrix $\boldsymbol{A} \in \mathbb{R}^{N \times N}$, where $N$ is the number of tokens in the input, i.e. the context length. Concretely, each transformer layer in the model first projects its input $\boldsymbol{X} \in \mathbb{R}^{N \times D}$, $D$ being the embedding size, to a query, key, and value matrix $\boldsymbol{Q}, \boldsymbol{K}, \boldsymbol{V} \in \mathbb{R}^{N \times d}$ using learnt weight matrices $\boldsymbol{W}_q, \boldsymbol{W}_k, \boldsymbol{W}_v \in \mathbb{R}^{D \times d}$, respectively, for each of the $H$ different heads with dimension $d$. Afterwards, the query and key matrices also receive a position encoding. In this work, we assume RoPE (Su et al., 2023) which right-multiplies each query and key $i$ with rope matrix $\boldsymbol{R}_i$. Hence, each row $i$ in $\boldsymbol{Q}, \boldsymbol{K}, \boldsymbol{V}$ is defined as $\boldsymbol{q}_i = \boldsymbol{x}_i \boldsymbol{W}_q \boldsymbol{R}_i$, $\boldsymbol{k}_i = \boldsymbol{x}_i \boldsymbol{W}_k \boldsymbol{R}_i$, and $\boldsymbol{v}_i = \boldsymbol{x}_i \boldsymbol{W}_v$, respectively. Finally, it computes the attention as the matrix product

$$\boldsymbol{A}\boldsymbol{V}, \text{ where } \boldsymbol{A} = \text{softmax}\left(\frac{\boldsymbol{Q}\boldsymbol{K}^t - \infty \boldsymbol{M}}{\sqrt{d}}\right).$$

In prevalent text generating LLMs, which only consists of decoders, $\boldsymbol{M} \in \{0, 1\}^{N \times N}$ is an upper-triangular matrix with only 1's in the upper triangle. Effectively, each element $A_{i,j}$ in $\boldsymbol{A}$ is defined as

$$A_{i,j} = \frac{\exp\left(\boldsymbol{q}_i \boldsymbol{k}_j^t / \sqrt{d}\right)}{\sum_{\ell=1}^{i} \exp\left(\boldsymbol{q}_i \boldsymbol{k}_\ell^t / \sqrt{d}\right)} \text{ if } j \leq i \text{ and } 0 \text{ otherwise.}$$

The attention matrix $\boldsymbol{A}$ has a tendency to be sparse, which means that many elements in $\boldsymbol{A}$ are sufficiently close to zero that they don't matter in the attention output $\boldsymbol{O} = \boldsymbol{A}\boldsymbol{V}$. Concretely, let the

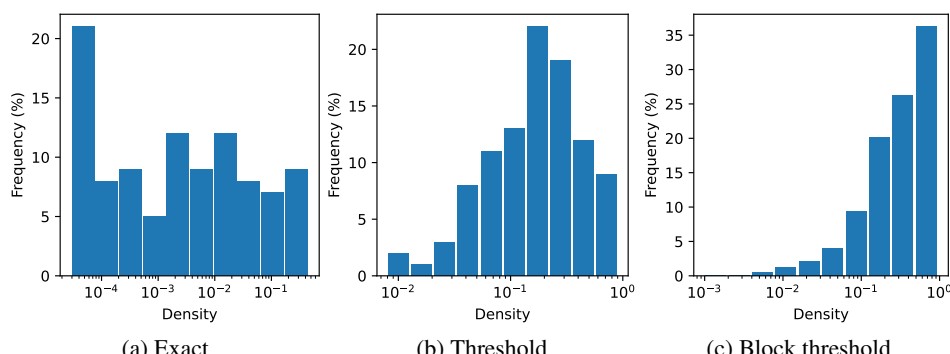

(a) Exact          (b) Threshold          (c) Block threshold

Figure 2: Distribution of attention matrix sparsity among all heads in LLAMA-3-8B-INSTRUCT-1048K on a prompt of 124k tokens using different criterion. The exact sparsity (a) is compared with the sparsity found the threshold criterion (b) and to threshold criterion when it is applied to the maximum value over $64 \times 64$ blocks (c).

attention mask $M$ mark all negligible elements in $A$, i.e. $\sum_{k:M_{i,k}=1} |A_{i,k}| < \varepsilon$ for some $\varepsilon > 0$, and approximate $O$ by $O'$ as $O'_{i,j} = \sum_{k:M_{i,k}=0} A_{i,k} V_{k,j}$, then

$$|O_{i,j} - O'_{i,j}| < \sum_{k:M_{i,k}=1} |A_{i,k}| \max_{1 \leq \ell \leq N} |V_{\ell,j}| \leq \varepsilon \|V\|_\infty \tag{1}$$

Hence, the estimate can be made arbitrarily accurate by taking $\varepsilon$ small enough.

Figure 2a shows the sparsity distribution among all heads in LLAMA-3-8B-INSTRUCT-1048K. A more practical method of measuring sparsity is counting or estimating the number of elements in $A$ that are below the threshold $\varepsilon/N$, because with only those elements left out, the same attention output error bound applies. This threshold condition, however, is less tight, which causes a large proportion of sparsity to go unnoticed, as illustrated by Figure 2b.

## 3 SPARSE PATTERN DECOMPOSITION

The sparsity of $A$ is not random. Indeed, several patterns emerge in $A$, which can be used to infer where the non-zeros are. Earlier work observed the $\Lambda$-shape (Han et al., 2024), vertical-slash (Jiang et al., 2024), block-sparse, and query-aware (Tang et al., 2024) patterns. We believe that the most relevant patterns can be captured by a decomposition of $A$ in three matrices $A_s, A_v, A_h \in \mathbb{R}^{N \times N}$ such that $A \approx A_s \odot A_v \odot A_h$, where

- $A_s$ is Toeplitz, meaning that it has constant diagonals and thus exhibits a slash-pattern,
- $A_v$ has constant columns and thus exhibits a vertical pattern, and
- $A_h$ has constant rows and thus exhibits a horizontal pattern.

We use the decomposition to find the non-zeros in $A$ by merely comparing their element-wise product with a threshold value, i.e. we set $M_{i,j} = 1$ for the elements for which $A_{i,j}^{(s)} A_{i,j}^{(v)} A_{i,j}^{(h)} < \varepsilon/N$, as is illustrated in Figure 3.

### 3.1 FINDING VERTICAL AND SLASH COMPONENTS

Jiang et al. (2024) observed that the majority of attention matrices have a vertical-slash pattern. Specifically, most attention matrices contain only a small number of columns that have many more non-zero elements than the other columns, causing a distinct vertical pattern in the visualisation of the matrix. The key tokens involved get high attention regardless of the query-token or the relative distance to it. Similarly, those matrices with a slash pattern show distinct diagonals in their visualization. Since all elements on the same diagonal have the same relative position while query

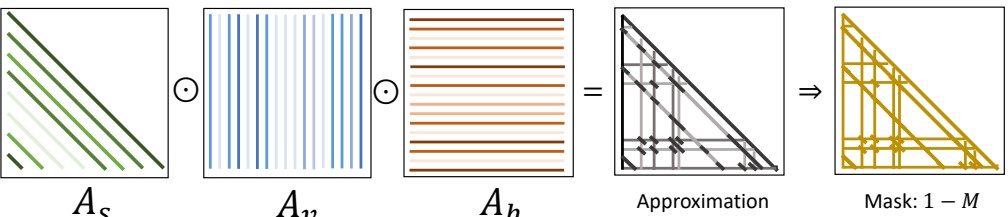

Figure 3: From decomposition to mask. The mask is computed from the element-wise product.

and key tokens are different every time, mainly the position encoding, i.e. RoPE, can be responsible for this. Our main idea is to predict such attention patterns with a linear model.

The key idea is to use a linear least-squares approximation of

$$\log \boldsymbol{A}_{ij} = \boldsymbol{q}_i \boldsymbol{k}_j^t / \sqrt{d} - \log \sum_{\ell=1}^{i} \exp\left(\boldsymbol{q}_i \boldsymbol{k}_\ell^t / \sqrt{d}\right). \tag{2}$$

As linear model, we could use $\boldsymbol{r}_{j-i}\boldsymbol{\alpha} + \boldsymbol{x}_j \boldsymbol{W}_k \boldsymbol{\kappa}$ with model parameters $\boldsymbol{\alpha}, \boldsymbol{\kappa} \in \mathbb{R}^d$ and our final decomposition would become $\boldsymbol{A}_{i,j}^{(s)} = \exp\left(\boldsymbol{r}_{j-i}\boldsymbol{\alpha}\right)$, $\boldsymbol{A}_{i,j}^{(v)} = \exp\left(\boldsymbol{x}_j \boldsymbol{W}_k \boldsymbol{\kappa}\right)$, and $\boldsymbol{A}_{i,j}^{(h)} = 1$. The $\boldsymbol{r}_{j-i}$ row-vector contains the RoPE coefficients

$$\boldsymbol{r}_{j-i} = \left(\cos\left((j-i)\Theta\right)\ \sin\left((j-i)\Theta\right)\right) \text{ where } \Theta = \left(\theta^{\frac{0}{d/2}}\ \dots\ \theta^{\frac{d/2-1}{d/2}}\right).$$

However, directly computing $\log \sum_{\ell=1}^{i} \exp\left(\boldsymbol{q}_i \boldsymbol{k}_\ell^t / \sqrt{d}\right)$ takes too much time, while ignoring this term degrades the quality of the approximation. Indeed, the row-average of $\boldsymbol{q}_i \boldsymbol{k}_j^t$ over all $j \leq i$ varies significantly between rows $i$, which effectively causes many outliers in the least-squares regression problem. Furthermore, elements $\boldsymbol{r}_{j-i}\boldsymbol{\alpha}$ and $\boldsymbol{x}_j \boldsymbol{W}_k \boldsymbol{\kappa}$ may be far way from zero, which degrades the approximation quality of $\boldsymbol{A}_{ij}^{(s)}$ and $\boldsymbol{A}_{ij}^{(v)}$ even more.

In particular, consider an arbitrary matrix $\boldsymbol{A}$ and its least-squares approximation $\boldsymbol{B}$, i.e. $||\boldsymbol{A} - \boldsymbol{B}||_F < \varepsilon$ for some $\varepsilon$. Then if we write $e^{\boldsymbol{A}}$ and $e^{\boldsymbol{B}}$ as the element-wise application of $\boldsymbol{A}$ and $\boldsymbol{B}$ to the exponential function, respectively, then we can bound $||e^{\boldsymbol{A}} - e^{\boldsymbol{B}}||_F$ by the use of a first-order Taylor expansion as

$$||e^{\boldsymbol{A}} - e^{\boldsymbol{B}}||_F = ||(\boldsymbol{A} - \boldsymbol{B})(1 + \frac{1}{2}(\boldsymbol{A} + \boldsymbol{B}) \odot mXi)||_F \leq \varepsilon ||1 + \frac{1}{2}(\boldsymbol{A} + \boldsymbol{B}) \odot mXi||_F$$

for some matrix $\boldsymbol{\Xi}$ with elements $|\Xi_{i,j}| < max(|A_{i,j}|, |B_{i,j}|)$. Hence, the smaller the elements of $\boldsymbol{A}$ and $\boldsymbol{B}$ are, the better the resemblance of $e^{\boldsymbol{A}}$ and $e^{\boldsymbol{B}}$ will be.

To tackle these issues, we center the linear model around zero by subtracting averages from each coefficient. Concretely, for each row $i$ we subtract the average RoPE value $\overline{\boldsymbol{r}}_i = \frac{1}{i}\sum_{j=1}^{i} \boldsymbol{r}_{j-i}$ from $\boldsymbol{r}_{j-i}$, the average unroped $K$ value $\overline{\boldsymbol{XW}_{ki}} = \frac{1}{i}\sum_{j=1}^{i} \boldsymbol{x}_j \boldsymbol{W}_k$ from $\boldsymbol{x}_j \boldsymbol{W}_k$, and the average $\boldsymbol{\mu}_i = \frac{1}{i}\sum_{j=1}^{i} \boldsymbol{q}_i \boldsymbol{k}_j^t / \sqrt{d}$ from $\boldsymbol{q}_i \boldsymbol{k}_j^t / \sqrt{d}$. The final linear model becomes

$$\left(\boldsymbol{r}_{j-i} - \overline{\boldsymbol{r}}_i\right)\boldsymbol{\alpha} + \left(\boldsymbol{x}_j \boldsymbol{W}_k - \overline{\boldsymbol{XW}_{ki}}\right)\boldsymbol{\kappa} \approx \boldsymbol{q}_i \boldsymbol{k}_j^t / \sqrt{d} - \mu_i \tag{3}$$

To find $\boldsymbol{\alpha}$ and $\boldsymbol{\kappa}$, we compute a small number of the coefficients and solve it as a linear least-squares problem. Unfortunately, the problem is rank-deficient regardless of the number of coefficients we sample. An SVD-based algorithm can handle rank-deficiency and would only require a sample size that is a small multiple of $2d$, i.e. the combined dimensions of $\boldsymbol{\alpha}$ ad $\boldsymbol{\kappa}$, to get a solution that is still able to generalize over all positions in the context. However, it much faster to take about ten times more samples and to apply ridge-regression through addition of a regularisation term, because it allows the use of the the highly efficient normal-equations algorithm. Nevertheless, the averages $\overline{\boldsymbol{r}}_i, \overline{\boldsymbol{XW}_{ki}}$, and $\mu_i$ have to be computed over all $N(N+1)/2$ possible values. Fortunately, these can be computed for all rows simultaneously in $\mathcal{O}(Nd)$ time.

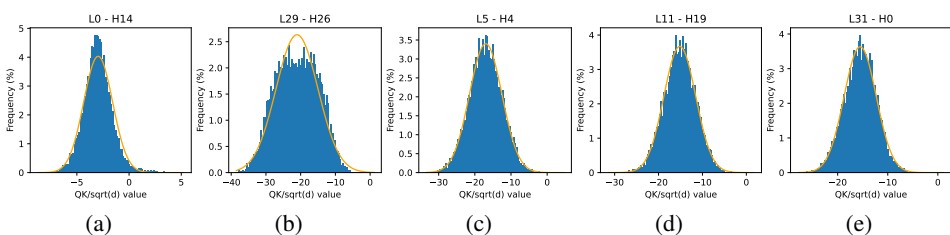

Figure 4: $QK^t/\sqrt{d}$ distributions from five randomly chosen heads from LLAMA-3-8B-INSTRUCT-1048K. Blue bars show the histogram of the values while the orange solid line shows the probability density function of the normal distribution with same mean and variance.

At this point, we have enough to identify the slash pattern $\boldsymbol{A}_s$ and the vertical pattern $\boldsymbol{A}_v$. Define vectors $\boldsymbol{s} \in \mathbb{R}^{2N-1}$ and $\boldsymbol{v}, \boldsymbol{\delta} \in \mathbb{R}^N$ as

$$s_{i-j} = (\boldsymbol{r}_{j-i} - \overline{\boldsymbol{r}}_N)\,\alpha \text{ for } j \le i \text{ and } -\infty \text{ otherwise} \tag{4}$$

$$\delta_i = (\overline{\boldsymbol{r}}_i - \overline{\boldsymbol{r}}_N)\,\boldsymbol{\alpha} \tag{5}$$

$$v_j = \left(\boldsymbol{x}_j \boldsymbol{W}_k - \overline{\boldsymbol{X}\boldsymbol{W}_{ki}}\right) \boldsymbol{\kappa} \tag{6}$$

and let elements from $\boldsymbol{A}_s$ be $A_{i,j}^{(s)} = e^{s_{i-j}}$ and elements from $\boldsymbol{A}_v$ be $A_{i,j}^{(v)} = e^{v_j}$, then we have

$$A_{i,j}^{(s)} A_{i,j}^{(v)} \approx \exp\left(\boldsymbol{q}_i \boldsymbol{k}_j^t/\sqrt{d} + \delta_i - \mu_i\right) \text{ for } j \le i \text{ and } 0 \text{ otherwise} \tag{7}$$

### 3.2 FINDING THE HORIZONTAL COMPONENT

Recall that our final goal is to approximate $\boldsymbol{A}$, but $\boldsymbol{A}_s \odot \boldsymbol{A}_v$ falls short by the term $\exp\left(-\delta_i + \mu_i - \log \sum_{\ell=1}^{i} \exp\left(\boldsymbol{q}_i \boldsymbol{k}_\ell^t/\sqrt{d}\right)\right)$. What remains to be found is

$$\nu_i \approx log \sum_{\ell=1}^{i} \exp\left(\boldsymbol{q}_i \boldsymbol{k}_\ell^t/\sqrt{d} - \mu_i\right), \tag{8}$$

so that $s_{i-j} + v_j - \delta_i - \nu_i \approx \log A_{i,j}$. To estimate $\nu_i$, we rely on the statistical distribution of $\boldsymbol{q}_i \boldsymbol{k}_j^t$, which is observed to be almost normal as illustrated in Figure 4, except for the first few $j < S$, so-called sink tokens (Xiao et al., 2024b), and the last few tokens $j > i - T$, i.e. the $\Lambda$-shape (Han et al., 2024). Therefore, $\exp\left(\boldsymbol{q}_i \boldsymbol{k}_j^t\right)$ will be log-normally distributed for most $j$ s.t. $S < j < i - T$ and therefore has expected value $\mathbb{E}(\boldsymbol{q}_i \boldsymbol{K}^t) = \exp\left(\mu_i + \sigma_i^2/2\right)$. To make sure that the distribution is matched exactly for at least the last row of $\boldsymbol{Q}\boldsymbol{K}^t/\sqrt{d}$, we apply a correction factor $\zeta$.

$$\zeta = \mu_N + \sigma_N^2/2 - \log \sum_{j=S+1}^{N-T} \exp\left(\boldsymbol{q}_i \boldsymbol{k}_j^t/\sqrt{d}\right). \tag{9}$$

The definition of $\nu_i$ becomes

$$\nu_i = \log\left(\Lambda_i + (i - S - T)\exp\left(\sigma_i^2/2 - \zeta\right)\right), \tag{10}$$

where $\Lambda_i = \sum_{j=1}^{S} \exp\left(\boldsymbol{q}_i \boldsymbol{k}_j^t/\sqrt{d} - \mu_i\right) + \sum_{j=i-T+1}^{i} \exp\left(\boldsymbol{q}_i \boldsymbol{k}_j^t/\sqrt{d} - \mu_i\right)$ are the sums over the $\Lambda$-shape.

The horizontal pattern $\boldsymbol{A}_h$ can, thus, be described with a vector $\boldsymbol{h} \in \mathbb{R}^N$ as

$$h_i = -\delta_i - \nu_i. \tag{11}$$

Let the elements of $\boldsymbol{A}_h$ be $A_{i,j}^{(h)} = e^{h_i}$, we now have $\boldsymbol{A}_s \odot \boldsymbol{A}_v \odot \boldsymbol{A}_h \approx \boldsymbol{A}$ for *the interior elements of* $\boldsymbol{A}$. That is, $\boldsymbol{A}$ without the $\Lambda$-shape. After all, we don't need to approximate the elements $\boldsymbol{Q}\boldsymbol{K}^t/\sqrt{d}$ in $\Lambda$, because those have been computed exactly already and their distribution is so different from the values in the interior, that it degrades the quality of the linear least-squares fit.

As illustrated in Figure 5, the procedure is run for each attention head individually with its respective inputs $\boldsymbol{Q}$ and $\boldsymbol{K}$. From this input the $\boldsymbol{Q}\boldsymbol{K}^t/\sqrt{d}$ sample is taken, the row-wise mean and variance $\boldsymbol{\mu}$ and $\boldsymbol{\sigma}^2$ is computed, and the sink, main-diagonal, and last row are computed. The RoPE term $(\boldsymbol{r}_{j-i} - \overline{\boldsymbol{r}}_i)$ can even be computed before any prefill takes place. The unRoPEd $\boldsymbol{x}_j\boldsymbol{W}_k$ can be computed by undoing RoPE on $K$ or, to save computation, it can be taken from an earlier stage. The linear least-squares produces the slash and vertical pattern. The horizontal pattern is directly computed from the $\Lambda$-shape, $\boldsymbol{\mu}$ and $\boldsymbol{\sigma}^2$. The detailed algorithm can be found in the appendix as algorithm 1.

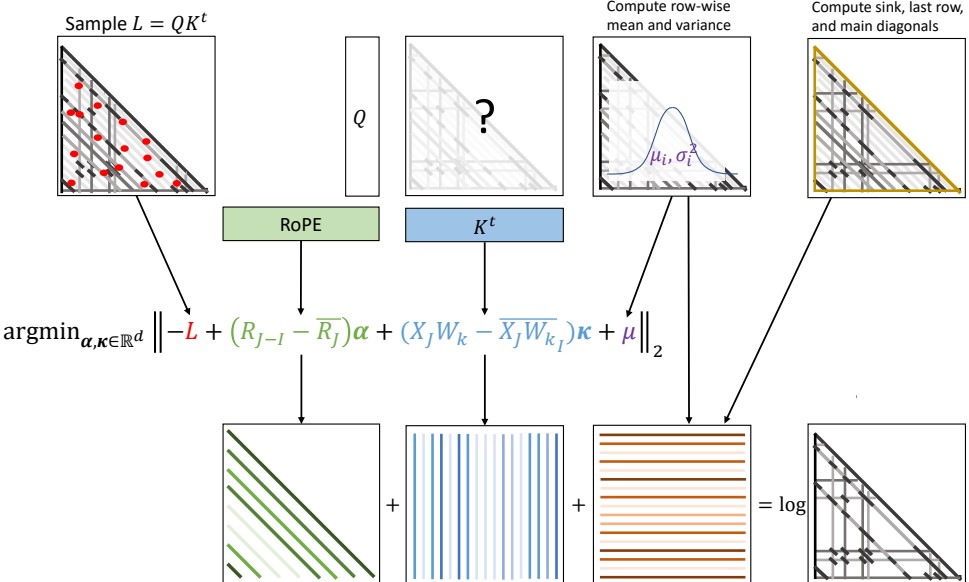

Figure 5: Overview of the steps in the decomposition algorithm

### 3.3 MASKING BY BLOCK-WISE COMPARISONS

To speed-up attention computation, the number of elements that are masked out from the attention matrix $\boldsymbol{A}$, must be maximized. However, masking out elements will introduce a numerical error, as is evident from equation (1), and only sufficiently many elements can be masked out if the error is allowed to be large enough. Unfortunately, it is unclear what the relation is between this error and the accuracy on an NLP task. The feed-forward layer in the transformer is capable of correcting numerical errors but this is not easy to model. Nevertheless, it is notationally convenient to use a threshold-based criterion $A_{i,j} < e^\tau = \varepsilon/N$ to distinguish zeros from non-zeros, because it relates to equation (1) which makes clear that $\varepsilon = 0$ selects all attention elements for computation while $\varepsilon \geq 1$ selects none.

The downside of using a threshold is that computational cost still grows super linearly because we observe that sparsity in most heads does not increase fast enough when the context length increases, as has been predicted mathematically by Deng et al. (2024). Alternatively, one can also fix the computational budget to something that increases linearly with the context length, while using the threshold to determine the budget at an intersect point. Since almost all attention matrices have log-normally distributed elements with parameters $\mu$ and $\sigma^2$, it is easy to convert between the two. Indeed, sparsity $p \in [0, 1]$, as the ratio of zero over non-zero elements, can statistically also be defined as a the probability that an element is below the threshold $\tau$

$$p = 2\frac{|\{A_{i,j} < e^\tau\}|}{N(N+1)} \approx \Phi((\tau - \mu)/\sigma),$$

where $\Phi$ is the cumulative density function of the standard normal distribution.

The decomposition provides an efficient way to estimate $\mu$ and $\sigma$ in quasi-linear time, because

$$\mu = \frac{2}{N(N+1)} \sum_{i=1}^{N} (is_i + (N-i+1)v_i + ih_i) \tag{12}$$

$$\sigma^2 + \mu^2 = \frac{2}{N(N+1)} \left( \sum_{i=1}^{N} \sum_{j=1}^{i} (s_{i-j+1} + v_h + h_j)^2 \right), \tag{13}$$

where the two convolutions required to compute $\sigma^2 + \mu^2$ have the dominant time complexity.

Although the decomposition approximates each individual element of the attention matrix, we will use the block-sparse attention kernel by Jiang et al. (2024) which requires that the mask is per $B \times B$ block with $B = 64$ and not element-wise. This changes the relation between $\tau$ and sparsity $p$ somewhat, because now each block will contain only $B$ independent random variables. Indeed, viewing the decomposition $s, v, h$ as three random vectors, their composition $s_{i-j} + vj + vi$ on a grid element $(i, j)$ is only independent from the other elements on its anti-diagonal. Considering this, we obtain

$$p_B \approx (\Phi((\tau_B - \mu)/\sigma))^B. \tag{14}$$

Note that $\tau_B = \tau + \log B$ is also different. In fact, it should coincide with the maximal error for each individual block of which there are only $N/B$. Finally, the decomposition is converted to the block structure by defining $v_i^{(B)} = \max_{0 \leq j < B}\{v_{iB+j}\}$, $h_i^{(B)} = \max_{0 \leq j < B}\{h_{iB+j}\}$, and $s_i^{(B)} = \frac{1}{2}(m_i + m_{i+1})$, where $m_i = \max_{0 \leq j < B}\{s_{iB+j}\}$. Blocking is a two-edged sword. One the hand, efficiency is gained by decreasing the mask size by $B^2$, while, on the other hand, sparsity is lost, as is show-cased by Figure 2c.

Another complication caused by blocking is that it changes the distribution of the attention and its decomposition differently. This distorts the threshold computation. We, therefore, compute B full rows of $A$ at a reference point $N_0$ and extrapolate the density to the entire matrix. This density is then converted to threshold $\tau_B$ using Equation 14.

In our experiments, with contexts going up to one million tokens, the quadratic-time algorithm that directly sets $M \in \{0, 1\}^{N/B \times N/B}$ as

$$M_{i,j} = \begin{cases} 1 & \text{if } s_{j-i}^{(B)} + v_j^{(B)} + h_i^{(B)} < \tau_B \\ 0 & \text{otherwise,} \end{cases}$$

is still faster than an alternative linear-time algorithm that selects elements by iterating over the dominant diagonals, columns, and rows. The linear-time algorithm is detailed by Algorithm 2 in the appendix.

## 4 EXPERIMENTS

In this section, we present an extensive overview of our experiments in order to evaluate SPARSE-SKELETON in terms of accuracy and speed-ups. We use a machine with 80G memory GPU to run our experiments.

**Implementation details.** The method is integrated in vLLM (Kwon et al., 2023) and is executed during the prefill phase of inference. Moreover, we implemented custom CUDA kernels to handle the critical part of the decomposition method. Non-crucial parts are computed directly relying on PyTorch library. The final sparse attention is computed with a custom Triton kernel (Tillet et al., 2019) which computes the attention at a block granularity. Finally, for the decode part, Flashattention (Dao, 2024) is used with a dense KV cache.

**Models.** We tested our solution with the long context capable LLAMA-3-8B-INSTRUCT-1048K.

**Benchmarks.** We use two benchmarks to evaluate our method: $\infty$ Bench (Zhang et al., 2024) and PG-19 (Rae et al., 2020).

**Baselines.** We compared SPARSESKELETON against three others techniques: dense attention computation with FlashAttention (Dao, 2024) ; MInference (Jiang et al., 2024) ; and FlexPrefill (Lai

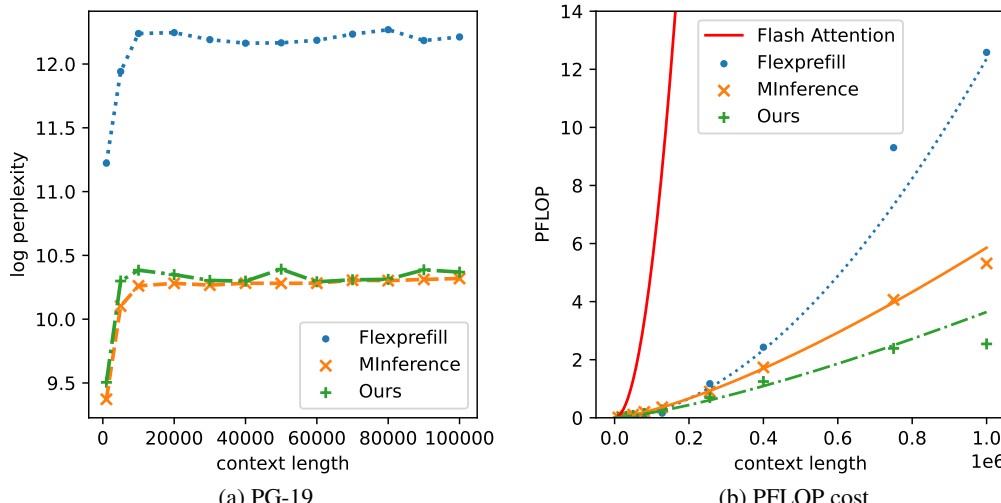

Figure 6: (a) PG-19 log perplexity result and (b) Sparse attention cost in PFLOP comparison between each prefill method

et al., 2025). For those three methods, we use the same dense decoding method in order to highlight only the impact of modified prefill phase.

We recorded perplexity on the PG-19 dataset, ranging from contexts of 1K to 100K tokens. As Figure 6a shows, the perplexity remains low, while we significantly reduce the number of FLOP computed during attention, as show in Figure 6b.

Subsequently, we record accuracy results on ∞bench without applying a time-budget limit but with a fixed threshold $\tau_B = \log \varepsilon B / N$ where $B = 64$ is the block size and $\varepsilon = 0.8$ the error tolerance. The results are listed in Table 1. Results with the time-budget limited mask are listed in Table **??** in the Appendix.

Table 1: Accuracy comparison on tested models and methods for different task of Infinite Bench ($N_{limit} = 262K$)

| METHOD | En.Sum | En.QA | En.MC | En.Dia | Zh.QA | Code.Debug | Math.Find | Retr.PassKey | Retr.Number | Retr.KV |
|---|---|---|---|---|---|---|---|---|---|---|
| LLama3-8B-1024k | 18.86 | **12.55** | 62.44 | 0.5 | 9.50 | 24.11 | 17.71 | **100** | **100** | 7.20 |
| MInference | 19.21 | 12.29 | **63.70** | 4.00 | 9.47 | 22.59 | **23.14** | **100** | **100** | 10.20 |
| FlexPrefill | 18.40 | 11.66 | 56.77 | 1.00 | 9.95 | 21.07 | 12.57 | **100** | **100** | 8.60 |
| Ours ($\epsilon = 0.8$) | **19.99** | 12.21 | 60.70 | **4.50** | **12.44** | **28.93** | 20.00 | **100** | 99.15 | 1.40 |
| Qwen2.5-14B-Instruct-1M | 22.22 | **13.19** | **79.91** | **39.50** | 9.99 | 45.82 | 35.43 | **100.00** | **100.00** | **92.20** |
| MInference | 21.98 | 9.03 | 77.73 | 21.00 | 10.18 | 48.86 | 34.00 | **100.00** | **100.00** | **92.20** |
| FlexPrefill | **22.72** | 8.84 | 78.60 | 18.50 | 9.57 | 37.47 | 42.29 | **100.00** | 99.49 | 80.60 |
| Ours (Budget Limit) | 22.62 | 12.81 | 69.43 | 21.50 | **10.41** | **50.38** | **42.86** | **100.00** | **100.00** | 80.45 |

## 5 RELATED WORK

Sun et al. (2025a) and Sun et al. (2025b) provide an excellent introduction and overview of sparse attention in their survey studies. In summary, the relevant works can be systematically categorized into three principal directions: (i) training-free methods employing plug-and-play mask predictor, (ii) post-training methods based on additional sparse mask learning, such as Gao et al. (2025b;a); Xiao et al. (2024a), and (iii) architectural approaches that incorporate native masked sparse attention mechanisms, an example solution is Yuan et al. (2025).

In brief, sparse attention concerns both training and inference, but the method is roughly the same, namely to avoid computing all elements in the attention matrix. Our method is training-free, it only

concerns the prefill stage of inference, it dynamically adapts to the input, and it uses block-sparse accelerated kernels. Similarly categorized are Jiang et al. (2024); Lai et al. (2025); Sahni et al. (2025); Zhang et al.; Peng et al. (2025). Yet, all solutions except ours and Sahni et al. (2025) need an extensive offline phase to find optimal pattern configurations.

Prior work relies on various patterns to approximate the prefill attention computation as effectively as possible. Early methods propose to compute attention in smaller localized attention windows while retaining the computation of the first sink tokens exploiting the $\Lambda$-shape pattern (Xiao et al., 2024b). Later methods introduce the vertical-slash pattern (Jiang et al., 2024). Lai et al. (2025) propose a fine-grained mask prediction method based on the divergence between the predicted distribution of attention scores and the true attention score distribution for sampled queries. The core idea is to support irregular masks through the block sparse pattern rather than relying solely on vertical-slash pattern, thereby further reducing computational cost while maintaining performance. More recent method tends to rely on a block sparse patterns, being more accelerator friendly. Sahni et al. (2025) propose to find attention blocks by estimating attention on the anti-diagonal which intersects both slashes and verticals. Alternatively, Ji et al. (2025) utilize quantisation as estimation method for blocks of attention.

## 6 DISCUSSION

Although we demonstrated that our method could significantly reduce the number of floating-point operations during the sparse attention computation, further implementation efforts are still needed to achieve end-to-end speedups. Especially, we need to develop high-efficiency parallel computation kernels for all the steps of the sparse pattern decomposition depicted in Section 3. Fortunately, there remains quite a bit of potential speed-up, which future work could address, possibly by also exploiting NPU technology like ASCEND coupled with VLLM-MINDSPORE (MindSpore (2025)).

## 7 CONCLUSION

This work decomposes an approximate attention matrix $A$ in three matrices: one with constant diagonals, one with constant columns, and one with constant rows. This decomposition captures most relevant emergent sparsity patterns in $A$, while it also readily provides a selection method that only relies on element or block-wise operations.

Using the LLAMA-3-8B-INSTRUCT-1048K model from Grattafiori et al. (2024), we observe more than 99.5% sparsity on average for inputs of one million tokens. In smaller context, 128K tokens, we obtain 97% sparsity ; and 89% sparsity for 10K tokens. Although, frontier LLMs seem to move away from MHA or GQA by using linear-attention methods (Qwen Team, 2025), our method provides an easy on-line way to accelerate the majority of LLMs that still use quadratic attention methods.

This exploratory work allows better sparsity. Even though, the number of FLOP used to compute the decomposition is dwarfed by the number of FLOP in the final block-sparse attention. Despite having developed GPU kernels for most time-critical parts, the time required by the decomposition is still considerable. Future work can exploit the absence of any input in the RoPE term in Equation 3. This absence suggests that the slash pattern is independent from the input. Future work could therefore attempt to extend the slash and vertical components during the inference's decode stage by using the $\alpha$ and $\kappa$ obtained during prefill.

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

# A APPENDIX

## A.1 ABLATION STUDY

**Algorithm 1:** Find decomposition

**Data:** The (RoPEd) query and key states $Q$ and $K$

**Result:** The decomposition $s, v, h \in \mathbb{R}^N$ and $\Lambda$-shape $sink, diags$

```
/* Uniform random sample of the interior of QK^t */
```
1 Sample $\mathcal{I} = \{(I_1, J_1), \ldots, (I_{80d}, J_{80d})\}$ where $I_i = S + T + \max(X_i, Y_i)$ and $J_i = S + \min(X_i, Y_i)$ with $X_i, Y_i$ uniformly randomly sampled from $[1..N - S - T]$

```
// Undo position encoding of K
```
2 $x_j W_k \leftarrow k_j R_j$ for all $S < j \leq N - T$

```
// QK row statistics
```
3 $\mu_i \leftarrow \frac{1}{\sqrt{d}i} q_i \sum_{j=1}^{i} k_j^t$ for all $1 \leq i \leq N$

4 $\sigma_i^2 \leftarrow \frac{1}{di} q_i \left( \sum_{j=1}^{i} k_j^t k_j \right) q_i^t - \mu_i^2$

```
// Λ-shape and last row of QK
```
5 $sink_{i,j} \leftarrow q_i k_j^t / \sqrt{d}$ for all $i, j$ s.t. $S + T < i \leq N$ and $1 \leq j \leq S$

6 $lastrow_j \leftarrow q_N k_j^t / \sqrt{d}$ for all $j$ s.t. $S < j \leq N - T$

7 $diags_{i,j} \leftarrow q_i k_{j+i-S-T}^t / \sqrt{d}$ for all $i, j$ s.t. $S + T < i \leq N$ and $1 \leq j \leq T$

8 $lambda_i \leftarrow \sum_{j=1}^{S} \exp(sink_{i,j} - \mu_i) + \sum_{j=1}^{T} \exp(diags_{i,j} - \mu_i)$

```
// Vertical and slash patterns
```
9 $\overline{r}_i \leftarrow \frac{1}{i - S - T} \sum_{j=S+1}^{i-T} r_{j-i}$ for all $i$ s.t. $S + T < i \leq N$

10 $\overline{XW_k}_i \leftarrow \frac{1}{i - S - T} \sum_{j=S+1}^{i-T} x_j W_k$ for all $i$ s.t. $S + T < i \leq N$

11 $A \leftarrow \left( (r_{J-I} - \overline{r}_I) \quad (x_J W_k - \overline{XW_k}_I) \right)$

12 $b_i \leftarrow q_i k_j^t / \sqrt{d} - \mu_i$ for all $(i, j) \in \mathcal{I}$

13 Apply normal-equations to solve $\arg\min_{\alpha, \kappa \in \mathbb{R}^d} || \left( \begin{smallmatrix} A \\ \lambda I \end{smallmatrix} \right) \left( \begin{smallmatrix} \alpha \\ \kappa \end{smallmatrix} \right) - \left( \begin{smallmatrix} b \\ 0 \end{smallmatrix} \right) ||_2$ for $\lambda = \varepsilon ||A||_F / 2$

14 $s_j \leftarrow (r_{j-N} - \overline{r}_N) \alpha$

15 $v_j \leftarrow (x_j W_k - \overline{XW_k}_N) \kappa$

```
// Horizontal pattern
```
16 $\zeta \leftarrow \mu_N + \sigma_N^2 / 2 - \log \sum_{j=S+1}^{N-T} \exp(lastrow_j)$

17 $\nu_i \leftarrow \log(lambda_i + (i - S - T) \exp(\sigma_i^2/2 - \zeta))$

18 $\delta_i \leftarrow \frac{1}{i - S - T} \sum_{j=S}^{i-T} s_{N-j+1}$

19 $h_i \leftarrow -\delta_i - \nu_i$

20 **return** $s, v, h, sink, diags$

Table 2: upper bound of the FLOP cost of decomposition method for each Attention head. See Algorithm 1 for detailed operation.

| STEP | FLOPS Cost |
|---|---|
| QK random sample (l.1) | $sample \times d^2$ |
| UnRoPEd K (l.2) | $2 \times d \times N$ |
| Row-wise mean (l.3) | $4 \times d \times N$ |
| Row-wise variance (l.4) | $2 \times d^2 \times N$ |
| S sinks (l.5) | $2 \times S \times d \times N$ |
| Last row (l.6) | $2 \times d \times N$ |
| T diagonals (l.7) | $2 \times T \times d \times N$ |
| Lambda (l.8) | $4 \times (S + T) \times N$ |
| RoPE average (l.9) | $4 \times d \times N$ |
| UnRoPEd K average (l.10) | $4 \times d \times N$ |
| A (l.11) | $4 \times d \times sample$ |
| b (l.12) | $sample$ |
| Least-square solving with A and b (l.13) | $2/3 d^3 + 2(sample + d)d^2$ |
| Slash component (l.14) | $3 \times d \times N$ |
| Vertical component (l.15) | $3 \times d \times N$ |
| Horizontal component (l.16-19) | $10 \times N$ |
| Total | $((2d + 2(S + T) + 22)d + 4(S + T) + 10) \times N + 8/3 d^3 + (3d^2 + 4d + 1) \times sample$ |

Table 3: Breakdown of FLOPS cost with LLAMA-3-8B-INSTRUCT-1048K: $d = 128, sample = 10k, S = 1, T = 100$. Used prompt is identical to figure 6.b.

| Context length | 10K | 20K | 30K | 50K | 80K | 128K | 256K | 400K | 750K | 1M |
|---|---|---|---|---|---|---|---|---|---|---|
| Decomposition (%) | 13.71 | 6.40 | 4.36 | 4.51 | 3.53 | 3.06 | 2.64 | 2.15 | 2.00 | 2.02 |
| Sparse Attention (%) | 86.29 | 93.6 | 95.64 | 95.49 | 96.47 | 96.94 | 97.36 | 97.85 | 98.00 | 97.98 |

---

**Algorithm 2:** Find mask

**Data:** The decomposition $s, v, h \in \mathbb{R}^N$ and threshold value $\tau$
**Result:** An index list $\mathcal{I} = \{(i_1, j_1), \ldots, (i_n, j_n)\}$
1 $\mathcal{I} \leftarrow \{\}$;
2 **foreach** $j$ *in order of decreasing* $argsort(v_j)$ **do**
3 $\quad$ S $\leftarrow \{(i,j) \mid v_j + s_{j-i} + h_i > \tau\}$;
4 $\quad$ **if** $3|S| < N - S - T - j + 1$ **then** break;
5 $\quad$ $\mathcal{I} \leftarrow \mathcal{I} \cup S$;
6 **end**
7 **foreach** $j$ *in order of decreasing* $argsort(s_j)$ **do**
8 $\quad$ S $\leftarrow \{(i,j) \mid v_j + s_{j-i} + h_i > \tau\}$;
9 $\quad$ **if** $3|S| < j$ **then** break;
10 $\quad$ $\mathcal{I} \leftarrow \mathcal{I} \cup S$;
11 **end**
12 **foreach** $i$ *in order of decreasing* $argsort(h_i)$ **do**
13 $\quad$ S $\leftarrow \{(i,j) \mid v_j + s_{j-i} + h_i > \tau\}$;
14 $\quad$ **if** $3|S| < i$ **then** break;
15 $\quad$ $\mathcal{I} \leftarrow \mathcal{I} \cup S$;
16 **end**

---

