# OpenReview forum: "SparseSkeleton: Prefill sparse attention by decomposition"
_ICLR.cc/2026/Conference — Submitted to ICLR 2026_

### Official Review · Reviewer_ywap · 2025-10-30

**Soundness:** 3
**Presentation:** 3
**Contribution:** 2
**Rating:** 2
**Confidence:** 4

**Summary:**

The paper proposes SparseSkeleton, a training-free, online prefill sparse-attention method that decomposes the attention matrix into three factors: slash, vertical, and horizontal, and then builds a block-sparse mask by thresholding or by meeting a time-budget target. It supports MHA/GQA with RoPE without retraining and integrates into vLLM with custom kernels for the prefill phase. Experiments on PG-19 and ∞Bench with Llama-3-8B-Instruct-1048K show comparable accuracy to dense and prior sparse-prefill baselines (including MInference) while further reducing prefill PFLOPs.

**Strengths:**

1. The design is training-free, with no model modification.
2. The slash/vertical/horizontal factorization is simple, statistically motivated, and maps cleanly to block-sparse execution with either an error-threshold or budget control.
3. Authors integrated the method into vLLM with custom kernels for prefill, demonstrating the engineering practicality.

**Weaknesses:**

1.  Although well-packaged and more unified, the proposed method appears to be a reformulation and modest generalization of MInference. The core ideas of combining diagonal and vertical structures to sparsify prefill attention are already central to prior work. The “horizontal” term is a reasonable extension, but its essentialness is not convincingly demonstrated.
2. The paper emphasizes PFLOPs and mask density, but lacks the TTFT evaluation, which is important for real acceleration.
3. On ∞Bench, the method underperforms MInference on some retrieval-heavy tasks (e.g., Retrieval.KV: 1.40 vs 10.20), indicating possible over-pruning of long-range signals. This raises concerns about the robustness of the sparsity pattern across diverse workloads.
4. The paper lacks strong ablation studies to justify the necessity of all three components. It remains unclear how much the horizontal term contributes relative to slash + vertical (as in MInference), or whether the added complexity is always worth the compute.
5. Only evaluated on a single model, single scale. The method is only tested on LLaMA-3-8B-Instruct. It remains uncertain how well this scales to larger or smaller models (e.g., 1.7B, 14B, 32B).

**Questions:**

1. What is the marginal gain of the horizontal term? Could you provide an ablation where A_h is removed or replaced with a constant, and quantify its effect on FLOPs and accuracy?
2. What is the actual runtime breakdown and the e2e latency? Could you quantify the proportion of total prefill time spent in decomposition vs attention computation?

---

### Official Review · Reviewer_mqk9 · 2025-11-01

**Soundness:** 1
**Presentation:** 2
**Contribution:** 3
**Rating:** 2
**Confidence:** 4

**Summary:**

The paper proposes a new sparse attention mechanism based on online decomposition. It decomposes the standard attention computation into three components: *slash*, *vertical*, and *horizontal*. The product of these components is used to construct a block-sparse mask during prefill. Experiments validate the FLOPs and performance of the proposed method.

**Strengths:**

1. The proposed method is clearly formulated and well-explained. Alternative designs and approximations are discussed systematically.
2. The method demonstrates improved average performance and reduced FLOPs compared to existing sparse attention baselines across evaluated benchmarks.

**Weaknesses:**

1. Limited experiments: The paper lacks sufficient empirical validation. Although FLOP reduction is reported, no end-to-end wall-clock speedup is provided, leaving readers uncertain about the practical efficiency gains. Moreover, ablation studies analyzing the contribution of each component are missing. The paper also does not analyze the resulting sparse patterns in detail. The experiment section is the main weakness of the paper and dominates the reviewer's final decision.

2. Presentation: The discussion of the method is somewhat flat, without emphasizing the key principles or design motivations early on. This structure may make it difficult for readers to grasp the main ideas before diving into implementation details.

**Questions:**

1. What is the end-to-end wall-clock speedup achieved by the proposed method compared to dense attention?

2. In which types of tasks or data domains does the proposed method particularly excel compared to other sparse attention methods?

---

### Official Review · Reviewer_kgAn · 2025-11-03

**Soundness:** 2
**Presentation:** 3
**Contribution:** 2
**Rating:** 4
**Confidence:** 3

**Summary:**

The paper proposes SparseSkeleton, a training-free, online prefill sparsification scheme that factorizes the attention matrix into three interpretable components, slash (Toeplitz/diagonal), vertical (columnwise), and horizontal (rowwise), and forms a block-sparse prefill mask by thresholding the product of these factors. The method is designed for MHA/GQA with RoPE and is integrated into vLLM with custom CUDA/Triton kernels for prefill; decoding remains dense (FlashAttention). Experiments on Llama-3-8B-Instruct-1048K report similar PG-19 perplexity to dense and prior sparse-prefill baselines, reduced prefill PFLOPs, and ∞Bench results that are mixed across tasks.

**Strengths:**

The paper is clearly written and presents a well-motivated approach to sparse prefill attention through a simple and interpretable factorization into slash, vertical, and horizontal components. This decomposition is intuitive, mathematically grounded, and maps naturally to block-sparse execution. The method is training-free and integrates smoothly into vLLM with custom kernels, demonstrating good engineering practicality. Experimental results show reduced prefill PFLOPs with comparable perplexity to dense and prior sparse baselines, suggesting the approach can achieve meaningful compute savings without retraining. Overall, the paper's clarity, sound formulation, and practical integration are notable strengths.

**Weaknesses:**

While the method is well-formulated, its empirical validation is limited. The paper focuses heavily on theoretical FLOP reduction and mask sparsity but does not provide end-to-end runtime measurements such as TTFT or wall-clock latency. Without this evidence, it remains unclear whether the proposed method offers real-world acceleration beyond simulated efficiency gains. Additionally, the evaluation is restricted to a single model scale (Llama-3-8B-Instruct), leaving questions about generality and scalability unanswered.

The contribution over prior work, particularly MInference, is also somewhat incremental. The slash and vertical components largely replicate existing ideas, and while the horizontal factor is novel, its necessity and impact are not convincingly demonstrated. The absence of ablation studies isolating this term makes it difficult to assess how much it contributes to performance or sparsity improvements.

Finally, the paper's results on ∞Bench reveal inconsistent behavior, especially in retrieval-heavy tasks, where the method underperforms compared to prior approaches. This suggests potential over-pruning of long-range dependencies and raises concerns about robustness across task types. Together, these limitations weaken the overall empirical strength and make the contribution appear less substantial than it could be with more comprehensive experimentation and analysis.

**Questions:**

* What is the actual end-to-end speedup (TTFT or wall-clock prefill latency) compared to dense attention and existing sparse prefill baselines such as MInference?
* How much does the horizontal component contribute to accuracy and sparsity? Could you provide an ablation removing or simplifying this term?
* How does the runtime cost of the decomposition (computing slash, vertical, and horizontal factors) compare to the attention computation itself?
* Have you tested the method on different model scales or architectures to evaluate generality beyond Llama-3-8B?
* What factors explain the performance drop on retrieval-heavy tasks in ∞Bench, and could the mask generation be adjusted to preserve long-range attention in those cases?

---

### Author Response · Authors · 2025-11-20

**All Reviewers**

First of all, we want to thank you for spending your time to review our paper. We truly appreciate the opportunity to
improve our proposal from your feedback.

We want to clarify how the proposed solution fundamentally differs from SoTA as such as MInference: with Sparse Skeleton,
we are creating an approximated attention map using a fully dynamic method that has linear time complexity. We found the
need for 3 components to get a sufficiently accurate approximated attention map. The role of each component is to model a
particular feature of the attention head:

 - The slash pattern models the behavior of RoPE regardless of the context.

 - The vertical pattern models attention to tokens regardless of the query or its distance to it.

 - The horizontal pattern models the effect of the row-wise SoftMax, which relates all values of in each row together. In
particular, it models how attention is distributed between the Lambda-shape (sink and main diagonals) and the interior
(anything but the sink and main diagonals) of the attention map.


The major contribution differs from related works in that these 3 sparse patterns are composed multiplicatively, instead of
additively with MInference for instance. This is a fundamental difference, since in our case, non-selected verticals for instance
(the ones that are close to 0) will lead to very small values in the diagonals where they intersect. This allows capturing complex
sparse patterns. In MInference, for instance, this is not achievable. See figure 1 for an illustrated example. As this prompt is a
concatenation of multiple books, it causes a stair case pattern in the attention map of some heads. Whereas MInference is not
able to provide that particular coverage, our approach, especially thanks to the horizontal component Ah, is capable of doing so.

We also think this method can be generalized to other positional encoding schemes, in particular the ones that modify RoPE
for long context inference (i.e. DCA, YARN). Moreover, we do believe that our method has a more efficient way of sampling
information from each attention head: our method is not limited to a specific part of the attention matrix to determine the mask
(i.e. the last rows). We expect that it will reduce the amount of data extrapolation and reduce the overall error. In order to
compute final output from A.V, The current implementation uses Block mask but we plan to experiment with an element wise
mask in order to achieve further FLOP reduction.

---

> ### Author Response · Authors · 2025-11-20
>
> **Reviewer kgAN**
>
> 1. What is the actual end-to-end speedup (TTFT or wall-clock prefill latency) compared to dense attention and existing
> sparse prefill baselines such as MInference ?
> Considering TTFT, we performed a theoretical analysis of the FLOPS cost of our decomposition (Table 3) to put it in
> perspective with the sparse attention computation. We do have speedups compared to dense flash attention (10% on 128K input
> size). Methods are on the way to add wall clock speedups compared to MInference.
>
> 2. How much does the horizontal component contribute to accuracy and sparsity? Could you provide an ablation removing
> or simplifying this term?
> As we want an extensive ablation study depicting the importance of the Ah component inside the decomposition method, we
> do not expect it to be ready for rebuttal.
>
> 3. How does the runtime cost of the decomposition (computing slash, vertical, and horizontal factors) compare to the attention computation itself?
> The cost of decomposition is marginal ( 5% to 2%, N > 20K) compared to the cost of the sparse attention of the constructed
> mask (see Table 4 for a breakdown).
>
> 4. Have you tested the method on different model scales or architectures to evaluate generality beyond Llama-3-8B?
> 1
> Additional experiments are underway to demonstrate the robustness and reliability of the solution. they will be added in the
> final version. We already added Qwen2.5-14b-Instruct-1M (See Table 1).
>
> 5. What factors explain the performance drop on retrieval-heavy tasks in ∞Bench, and could the mask generation be adjusted
> to preserve long-range attention in those cases?
> We can not yet provide any adjustment for kv retrieval, in order to solve the accuracy breakdown. Kv-retrieval whose
> prompts consist of JSON dictionaries with GUIDs as keys and values is an input with low token diversity. In the derivation
> of our method, we have assumed that the distribution of the q-k inner-products is normal in order to estimate the row-wise
> SoftMax (see equation 8) for the horizontal component and the density of the final block mask (see equation 14). However,
> when the token diversity is low, the q-k distribution in the first few layers is quite far from normal. As a result, the quality of
> the block masks degrades in those layers which, by itself, degrades the final inference result. Nevertheless, we observed that this
> task is highly sensitive: some environment modification, such as changing VLLM and torch version, results in a 50% decrease in
> accuracy with flash-attention prefill baseline computation.
>
>
> **Reviewer mqk9**
>
> 1. What is the end-to-end wall-clock speedup achieved by the proposed method compared to dense attention?
> See Reviewer kgAN - 1.
>
> 2. In what types of tasks or data domains does the proposed method particularly excel compared to other sparse attention
> methods?
> We have observed that our method performs better in natural language processing tasks. This includes PG-19 benchmarks,
> or some specific tasks inside the ∞bench such as long text summarization, english or chinese question answering, code debugging.
> This topic is highly related to Reviewer kgAN - 5 discussion.
>
>
> **Reviewer ywap**
>
> 1. What is the marginal gain of the horizontal term? Could you provide an ablation where Ah is removed or replaced with
> a constant, and quantify its effect on FLOPs and accuracy?
> See Reviewer kgAN - 2. This new component allowed us to reduce the overall FLOPs by 2 compared to MInference while
> maintaining accuracy (See figure 6.a and 6.b). We believe that the attention head depicted in figure 1 is a good example of how
> the Ah component permits reducing the overall number of computed FLOPs as each row can have different FLOPs density.
>
> 2. What is the actual runtime breakdown and the e2e latency? Could you quantify the proportion of total prefill time spent
> in decomposition vs attention computation?
> See Reviewer kgAN - 1 and Reviewer kgAN - 3.

---

### Meta-Review · Area_Chair_6HCT · 2025-12-07

**Summary:**

While the paper introduces an interpretable and training-free sparsification method with a clean slash–vertical–horizontal factorization, all reviewers raised concerns about insufficient empirical validation, including missing end-to-end latency results, lack of ablations, and unclear incremental contribution relative to MInference.


The authors’ rebuttal provided conceptual clarifications about the decomposition and articulated why multiplicative composition differs from prior additive formulations. The response also partially explained failure cases (e.g., retrieval tasks where q–k distributions deviate from Gaussian assumptions) and provided limited runtime breakdown for decomposition overhead.


However, the rebuttal did not provide any new experiments, including: 1) No TTFT or wall-clock prefill latency. 2)No ablation studies of the horizontal component. 3)No additional analysis of sparse patterns. 4)No correction or improvement of retrieval-task underperformance. 5) No substantial multi-model-scale validation (the single Qwen result added is insufficient).


Since the core missing evidence was experimental rather than conceptual, clarifications alone could not resolve reviewers’ primary concerns. Thus, despite an interesting idea and a well-written rebuttal, the empirical gaps remain too significant, and the reviewers’ consensus continues to support rejection.

**Reviewer Concerns:**

**Concerns Partially Addressed**

1. Conceptual novelty and difference from MInference

    The authors clarified that:

    SparseSkeleton composes factors multiplicatively, not additively.

    This allows interactions that MInference cannot capture (e.g., suppressing diagonals via near-zero vertical terms).

    They provided conceptual examples (e.g., staircase patterns in concatenated books).

    ***Assessment***: Good clarification, but without ablations showing how much horizontal helps, the reviewers’ doubts about incremental novelty remain.



 2. Decomposition overhead vs. sparse attention cost

    The rebuttal states decomposition costs 2–5% of total prefill time for N > 20K.

    ***Assessment***:  This partially answers reviewers’ concerns, but without full TTFT or wall-clock measurement, it does not resolve the question of actual end-to-end speedup.


 3. Retrieval-task underperformance explanation

     Authors point out: (1) q–k distributions are non-Gaussian for GUID-based JSON inputs. (2) This breaks assumptions used for estimating SoftMax and density. (3) Even dense FlashAttention is unstable under environment changes.

    ***Assessment***: This gives a plausible theoretical explanation but does not provide mitigation, experiments, or mask-adjustment strategies, so the concern remains outstanding.


4. Additional model tested (Qwen2.5-14B)

   The authors added a table referencing Qwen2.5-14B-Instruct-1M.

   ***Assessment***: A single extra model is appreciated, but too limited to demonstrate scalability, and does not address concerns about robustness across multiple scales.


--------------------------------------------
**Concerns Not Addressed**

These remain the decisive reasons for recommending rejection.

 1. No end-to-end wall-clock speedup / TTFT evaluation

    All reviewers asked for it. The rebuttal says results are "on the way" but none were provided. This is the most critical missing evidence.

 2. No ablations of the horizontal term (Ah)

    Despite reviewer emphasis, authors explicitly said ablations would not be ready. Without this, reviewers cannot assess: (1) Necessity of Ah; (2) Incremental contribution over MInference; (3) Effect on FLOPs, accuracy, robustness

     Thus, the main novelty claim remains unsubstantiated.

3. No analysis of sparse pattern quality

    Reviewers asked for pattern visualization, robustness across layers, etc. Nothing new was added.

4. No improvements or adjustments for retrieval-heavy tasks

   The rebuttal acknowledges the degradation but does not propose a fix or an experimental evaluation of potential adjustments.

5. No expanded multi-model evaluation

   One additional model is insufficient; reviewers sought strong evidence across scales.

**Reviewer Scores:**

Reviewer kgAn asked strongly for TTFT, ablations, and multi-model tests. Since none were provided, their position is expected to be unchanged.



Reviewer mqk9 emphasized missing experiments and weak empirical validation. Rebuttal did not introduce new results; conceptual clarifications alone do not address core deficiencies.



Reviewer ywap focused heavily on lack of novelty justification and missing ablations. Since ablations were not provided and runtime results remain absent, no score increase is expected.

---

### Decision · Program_Chairs · 2026-01-26

Reject